# FAST TASK ADAPTATION FOR FEW-SHOT LEARNING

## ABSTRACT

Few-shot classification is a challenging task due to the scarcity of training examples for each class. The key lies in generalization of prior knowledge learned from large-scale base classes and fast adaptation of the classifier to novel classes. In this paper, we introduce a two-stage framework. In the first stage, we attempt to learn task-agnostic feature on base data with a novel Metric-Softmax loss. The Metric-Softmax loss is trained against the whole label set and learns more discriminative feature than episodic training. Besides, the Metric-Softmax classifier can be applied to base and novel classes in a consistent manner, which is critical for the generalizability of the learned feature. In the second stage, we design a task-adaptive transformation which adapts the classifier to each few-shot setting very fast within a few tuning epochs. Compared with existing fine-tuning scheme, the scarce examples of novel classes are exploited more effectively. Experiments show that our approach outperforms current state-of-the-arts by a large margin on the commonly used mini-ImageNet and CUB-200-2011 benchmarks.

## 1 INTRODUCTION

In recent years, deep learning models have achieved great success in many visual tasks such as image classification (Krizhevsky et al., 2012), object detection (Girshick, 2015) and semantic segmentation (Long et al., 2015). In general, training a deep network needs massively labeled instances which leads to expensive manual annotation cost. In contrast, humans can learn novel concepts from only one or a few examples. Analogously, few-shot learning aims to recognize unseen instances by accessing just a small number of labeled images in each class. Unfortunately, naive methods such as re-training or fine-tuning the model on novel data would severely suffer from overfitting and provide poor result (Finn et al., 2017). The major difficulty lies in the exceedingly limited data, which can hardly represent the class distribution.

In the task of few-shot classification, we are given three datasets, namely training set, support set and query set. The training set (also known as base dataset) normally contains large-scale labeled data for learning of prior knowledge. The support set and query set contain a small number of examples which are drawn from unseen classes disjoint with the training set. We are expected to classify the query images, leveraging the prior knowledge from the base training set and the limited cue from the support images. If the support set contains $k$ annotated images for each of $m$ unique classes, then this task is called $m$-way $k$-shot classification.

Previous works on few-shot learning can be roughly divided into two categories, namely meta-learning based and metric learning based. The former typically learns a meta-learner model on base data and generalizes it to novel unseen data. Usually, a recurrent neural network (Santoro et al., 2016) or long short-term memory network (Hochreiter & Schmidhuber, 1997) is utilized to learn a memory network to store knowledge. The latter attempts to learn a feature embedding where samples of the same class are closer to each other than samples of different classes. To make training on base data and inference on novel data consistent, a so-called episodic training strategy is widely employed (Vinyals et al., 2016; Snell et al., 2017; Sung et al., 2018). There are two limitations in metric learning based approaches. One is that episodic training only considers local sample similarity in current data batch. As shown in Snell et al. (2017), the limited number of classes per-training episode is harmful for discriminative feature learning, while simply increasing it leads to better performance. However, in a sampled episode, a large number of classes is infeasible due to limited GPU memory. The other limitation is that most existing methods only utilize the labeled

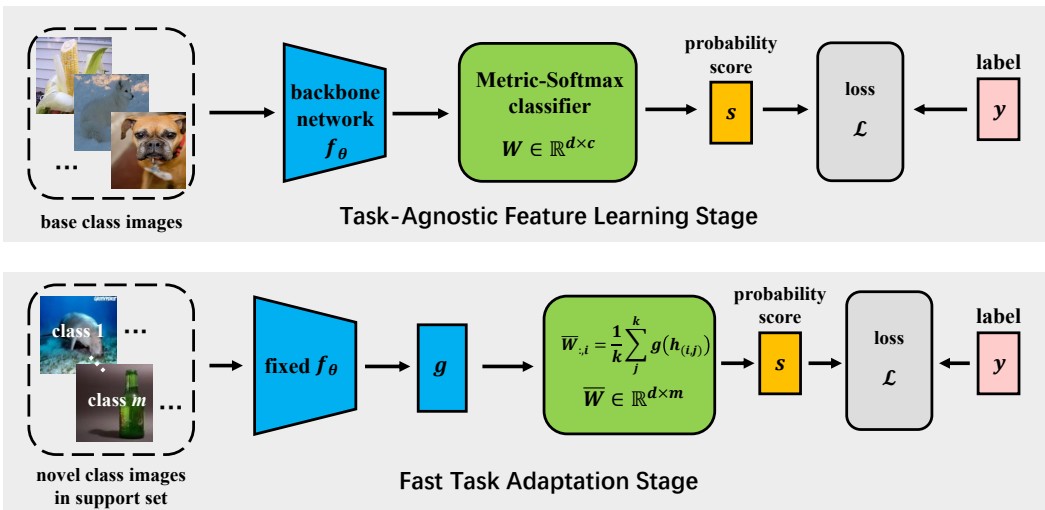

Figure 1: Overview of the proposed FTA framework, which is composed of two stages.

support images in novel classes as templates for feature similarity comparison. We argue that the support images could be better exploited to improve the performance.

In this work, we introduce a two-stage framework named Fast Task Adaption (FTA). As illustrated in Figure 1, the few-shot classification task is decomposed into two steps, i.e. task-agnostic feature learning on base data and adaptation of the classifier to each few-shot learning task. To learn more general and discriminative feature on base data, we propose a novel Metric-Softmax loss, which is an improvement of the Softmax loss. Specifically, we replace the score computing in softmax with a Gaussian kernel-based radial basis function. The Metric-Softmax loss is trained against the full base classes, overcoming the limitation of small number of classes in episodic training. Moreover, the Metric-Softmax classifier can be applied to base and novel classes in the same way. The consistency between training and inference is well preserved. After task-agnostic feature is learned, we design a novel task-adaptive transformation to adapt the classifier to current few-shot learning task. It is essentially an affine transformation applied to features of novel images. Notably, compared with fine-tuning re-initialized weights, it exhibits the desirable characteristics of fast convergence and superior performance. In this way, the aforementioned weaknesses of existing metric learning based methods are well addressed, leading to significant accuracy improvement. Experimental results on the mini-ImageNet and CUB-200-2011 datasets show that our approach consistently outperforms current state-of-the-art methods with respect to various backbone networks.

The main contributions of this paper can be summarized as follows.

- We propose the Metric-Softmax loss, which learns highly discriminative feature and ensures the consistency between training and inference as well.
- We design a novel task-adaptive transformation to adapt the classifier to novel classes efficiently and effectively.
- We improve few-shot classification significantly and achieve state-of-the-art performance on two common few-shot learning benchmarks.

## 2 RELATED WORK

In this section, we briefly review the previous approaches relevant to ours and current state-of-the-art methods we compare with in experiments.

**Meta-learning based methods** aim to learn task-agnostic knowledge from a sequence of similar tasks which have enough training data and apply it to new tasks for fast adaption. Ravi & Larochelle

(2017) utilize an LSTM-based network to control the parameter optimization process, which results in better generalization than SGD when training on a few labeled samples. Qiao et al. (2018) propose to adapt a pre-trained network to novel classes by directly predicting the parameters from the activations. MAML (Finn et al., 2017) learns sensitive and general initial parameters, which can fast adapt to a novel task with only one or a few steps of gradient-descent update. MTL (Sun et al., 2019) learns to adapt a deep neural network for few-shot learning tasks by learning scaling and shifting functions of weights for each task. Cai et al. (2018) use a contextual learner to predict the parameters of an embedding network for unlabeled data by using memory slots. Mishra et al. (2018) propose a meta-learner architecture that combines temporal convolutions and soft attention.

**Metric learning based methods** aim to learn a feature embedding that preserves the class neighborhood structure. Specifically, samples of the same class are closer than samples of different classes in the learned embedding space. Matching networks (Vinyals et al., 2016) combine attention and memory together by learning a network that maps a small labeled support set and an unlabeled example to its label. They also first introduce the episodic training strategy, where the training process mimics the test scenario based on support-query metric learning. Prototypical network (Snell et al., 2017) uses the mean feature of each class as its corresponding prototype representation to learn a similarity metric space. Relation network (Sung et al., 2018) learns a deep non-linear distance metric by considering the relation between query images and support images. Satorras & Estrach (2018) propose a graph neural network to learn the feature similarity between query and support set. Our approach belongs to this category. Compared with existing methods, our major improvements are the Metric-Softmax loss for generic feature learning and the task-adaptive transformation for fast task adaptation.

**Other methods.** Zhang et al. (2018) propose a GAN-based approach to help few-shot classifiers to learn sharper decision boundary. Si et al. (2019) propose a progressive cluster purification method for transductive few-shot learning. Li et al. (2019) propose a deep nearest neighbor neural network for few-shot learning. Chen et al. (2019) give a consistent comparative analysis of several representative few-shot classification algorithms and show that the depth of backbone network matters. Besides, they attempt to fine-tune a Softmax classifier on the support set. On the contrary, we first apply an affine transformation to features of the support images, and compute the class centroids of the transformed features as the classfier's weights. Concretely, we learn the affine transforming matrix rather than directly fine-tune the randomly initialized weights. Optimizing the transformation is computationally more stable even when one example per-class is available.

## 3 METHOD

As shown in Figure 1, the proposed FTA framework is composed of two stages, namely task-agnostic feature learning stage and fast task adaptation stage. In the first stage, we train the network on the base data with the proposed Metric-Softmax loss. After that, we fix the parameters of the network backbone and use it as a feature extractor. In the second stage, we learn a task-adaptive classifier, which is also a Metric-Softmax classifier with its weights replaced with affine transformed features of the support images. The affine transformation $g$ is named Task-Adaptive Transformation (TAT), which is learned on the scarce support set. With proper initialization, training of $g$ converges very fast within a few epochs. After training finishes, $g$ is applied to both support and query images.

In this section, we first elaborately describe the proposed Metric-Softmax loss. Since it is an improvement of the Softmax loss, we first give a brief review of the Softmax loss and reveal the issue of discrepancy between training and inference. Then we explain how to address this issue with the Metric-Softmax loss. Finally, we introduce the task-adaptive transformation for fast task adaptation.

### 3.1 SOFTMAX CLASSIFIER

Feature learning with the Softmax loss has achieved great success in many feature ranking tasks like person re-identification (Zhang et al., 2017) and face verification (Wen et al., 2016). Given an image $X$ of base classes, a backbone network $f$ is applied to project $X$ into feature space $h = f(X), h \in \mathbb{R}^d$. Then, the logits $z$ are computed by feeding $h$ into an affine transformation:

$$z = W^T h + b, \tag{1}$$

where $\boldsymbol{W} \in \mathbb{R}^{d \times c}$ and $\boldsymbol{b} \in \mathbb{R}^c$ are the transforming weights. $c$ is the number of classes in base data. The Softmax classifier predicts the probability score that the image belongs to class $i$ by

$$s_i = \text{softmax}(\boldsymbol{z})_i = \frac{\exp(z_i)}{\sum_j^c \exp(z_j)} \tag{2}$$

The Softmax loss is an assembly of the Softmax classifier and cross-entropy loss. Suppose the one-hot category label of the image is $\boldsymbol{y} \in \{0, 1\}^c$, the training cross-entropy loss is defined as

$$\mathcal{L} = -\sum_j^c y_j \log(s_j) \tag{3}$$

During inference, if a test image belongs to one of the training classes, its label can be predicted by

$$\hat{y} = \arg\max_i s_i \tag{4}$$

In this case, the scores in both training and inference are computed by an affine transformation and softmax function applied to the feature vector $\boldsymbol{h}$. However, in the context of few-shot learning, the query image for testing belongs to a novel class unseen in training. A common way to classify the novel query image is to compare its similarities to the few labeled support images, which share the same label space with the query image. Concretely, for $m$-way $k$-shot classification, we can extract features of the support images, and compute the mean feature for each class by

$$\bar{\boldsymbol{h}}_i = \frac{1}{k} \sum_j^k f(\boldsymbol{X}_j^i) \tag{5}$$

Here, $\boldsymbol{X}_j^i$ denotes the $j$-th image of class $i$ in the support set. The label of a query image $\boldsymbol{X}$ can be predicted by

$$\hat{y} = \arg\min_i ||f(\boldsymbol{X}) - \bar{\boldsymbol{h}}_i|| \tag{6}$$

Rather than an inner-product between $\boldsymbol{h}$ and column vectors of the weight matrix ($\boldsymbol{z}_i = \boldsymbol{W}_{:,i}^T \boldsymbol{h} + \boldsymbol{b}_i$) followed by softmax function in the previous case, the prediction is based on the euclidean distance between the query and support features. We argue that this discrepancy hurts the transferability of features pre-trained on base data to novel data.

## 3.2 METRIC-SOFTMAX CLASSIFIER

To eliminate this discrepancy, we improve the Softmax classifier by redefining the probability score calculating function. Specifically, we replace the $\exp(z_i)$ term in Equation 2 with a Gaussian kernel-based radial basis function, whose center is the column vector of a learnable weight matrix $\boldsymbol{W}_{:,i}, \boldsymbol{W} \in \mathbb{R}^{d \times c}$. Then the score can be computed by

$$s_i = \frac{\exp(-\alpha ||\boldsymbol{h} - \boldsymbol{W}_{:,i}||)}{\sum_j^c \exp(-\alpha ||\boldsymbol{h} - \boldsymbol{W}_{:,j}||)}, \tag{7}$$

where $\alpha \in \mathbb{R}, \alpha > 0$ is a hyper-parameter for scaling and $\boldsymbol{h}$ is an L2-normalized feature vector. In this way, it can be easily derived that $\arg\max_i s_i$ is equal to $\arg\min_i ||\boldsymbol{h} - \boldsymbol{W}_{:,i}||$. Here $\boldsymbol{W}_{:,i}$ can be interpreted as the centroid of class $i$ in the learned embedding space. That means the training procedure is essentially optimizing the euclidean distance-based similarity between images to their corresponding class centroids. During training, the cross-entropy loss is adopted as in the Softmax loss.

During inference, we replace $\boldsymbol{W}_{:,i}$ with the mean feature (see Equation 5) of class $i$ in the support set. Then classification of a query image is conducted according to Equation 4. Like episodic training, the consistency between training and inference is well preserved. And we easily enjoy the benefit of more general and discriminative feature learned with the cross-entropy loss. It is worth noting that the number of classes during inference on novel data is independent from the base dataset, which makes it flexible enough to adapt to different few-shot classification tasks.

## 3.3 Fast Task Adaptation

Although the feature learned on the large-scale base data is highly general and discriminative, it is unlikely to perfectly fit arbitrary novel classes. A proper adaptation to each few-shot task is beneficial. To leverage the valuable cue provided by the support images, a straightforward choice is to fine-tune a specific Softmax classifier as in Chen et al. (2019). However, even with the feature extractor being fixed, it is still difficult to train the classifier due to easy overfitting on small dataset. In contrast, we design a parameterized transformation function $g$ to adapt the general feature to current few-shot learning task,

$$h' = g(h) \tag{8}$$

Concretely, $g$ is simply a zero-offset affine transformation: $g(h) = M^T h, M \in \mathbb{R}^{d \times d}$. For $m$-way $k$-shot classification, we reconstruct an $m$-class Metric-Softmax classifier whose weight matrix $\bar{W} \in \mathbb{R}^{d \times m}$ is derived from the transformed features of the support images

$$\bar{W}_{:,i} = \frac{1}{k} \sum_j^k g(h_{(i,j)}) \tag{9}$$

Here $h_{(i,j)}$ denotes the feature of $j$-th image in class $i$. It is worth mentioning that both the transformed feature vector $g(h)$ and each column of the weight matrix $\bar{W}$ should be L2-normalized. And now, the computation of probability score is defined as

$$s_i = \frac{\exp(-\alpha || g(h) - \bar{W}_{:,i} ||)}{\sum_j^c \exp(-\alpha || g(h) - \bar{W}_{:,j} ||)} \tag{10}$$

The transformation $g$ is trained on all labeled images in the support set with the cross-entropy loss. To ease optimization, the transforming matrix $M$ is initialized with an identity matrix. After training of $g$ finishes, it is applied to both support and query images. And the final prediction of a query image $X$ is computed by

$$\hat{y} = \arg\min_i || g(f(X)) - \bar{W}_{:,i} || \tag{11}$$

## 4 Experiments

### 4.1 Datasets

We conduct few-shot classification experiments on two common benchmarks, namely mini-ImageNet (Vinyals et al., 2016) and CUB-200-2011 (Wah et al., 2011).

**mini-ImageNet** is first proposed by Vinyals et al. (2016) for few-shot classification evaluation. It consists of 100 classes and 600 images per class, which is a subset sampled from the ImageNet dataset (Deng et al., 2009). In our experiments, we follow the dataset splits proposed in Ravi & Larochelle (2017), which takes 64, 16 and 20 classes for training, validation and testing respectively.

**CUB-200-2011** is initially proposed for fine-grained classification of birds. It contains 200 categories and 11,788 images in total. We follow the evaluation protocol used in Hilliard et al. (2018), which randomly splits the dataset into 100, 50 and 50 classes for training, validation and testing.

### 4.2 Implementation Details

To make a fair comparison with existing works, we evaluate our method for three commonly used backbone networks, namely Conv-4 (Vinyals et al., 2016), ResNet-10 (Chen et al., 2019) and ResNet-12 (Oreshkin et al., 2018). Besides difference in depth and architecture, Conv-4 and ResNet-12 expect an input size of $84 \times 84$, while ResNet-10 takes $224 \times 224$ images as input, following the setting in existing works. In our experiments, we mainly focus on the 5-way 1-shot and 5-way 5-shot classification settings which are the most commonly used in existing works. The test episode contains 5 classes and each class contains 1 (or 5) support image(s) and 15 query images. For all experiments, we report the mean accuracy over 1200 randomly sampled episodes and the 95% confidence interval.

Table 1: Few-shot classification results on the test set of the mini-ImageNet dataset. $^*$Results reported by Chen et al. (2019).

| Method | Backbone | 5-way 1-shot | 5-way 5-shot |
|---|---|---|---|
| MatchingNet$^*$ (Vinyals et al., 2016) | Conv-4 | 48.14±0.78 | 63.48±0.66 |
| ProtoNet (Snell et al., 2017) | | 49.42±0.78 | 68.20±0.66 |
| MAML (Finn et al., 2017) | | 48.70±1.84 | 63.11±0.92 |
| RelationNet (Sung et al., 2018) | | 50.44±0.82 | 65.32±0.70 |
| PABN (Huang et al., 2019) | | 51.87±0.45 | 65.37±0.68 |
| Baseline++ (Chen et al., 2019) | | 48.24±0.75 | 66.43±0.63 |
| DN4 (Li et al., 2019) | | 51.24±0.74 | 71.02±0.64 |
| **FTA (Ours)** | | **52.13±0.53** | **80.35±0.41** |
| MetaGAN (Zhang et al., 2018) | ResNet-12 | 52.71±0.64 | 68.63±0.67 |
| SNAIL (Mishra et al., 2018) | | 55.71±0.99 | 68.88±0.92 |
| AdaResNet (Munkhdalai et al., 2018) | | 56.88±0.62 | 71.94±0.57 |
| TADAM (Oreshkin et al., 2018) | | 58.5±0.3 | 76.7±0.3 |
| MTL (Sun et al., 2019) | | 61.2±1.8 | 75.5±0.8 |
| Ravichandran et al. (2019) | | 59.00± - | 77.46± - |
| MetaOptNet-SVM (Lee et al., 2019) | | **62.64±0.61** | 78.63±0.46 |
| **FTA (Ours)** | | 58.03±0.58 | **80.73±0.44** |

Our implementation is based on PyTorch (Paszke et al., 2017). We apply the same data augmentation as Chen et al. (2019), including random cropping, horizontal flipping and color jittering in both feature learning and fast task adaptation stages. The Adam (Kingma & Ba, 2015) optimizer with $\epsilon = 10^{-3}$, $\beta_1 = 0.9$ and $\beta_2 = 0.999$ is used. In the feature learning stage, the backbone network is trained from scratch for 300 epochs in total with a batch size of 32. We decay the learning rate by a factor of 0.1 every 75 epochs. The scaling factor $\alpha$ used in Metric-Softmax is set to 15 for mini-ImageNet and 1 for CUB-200-2011. In the task adaptation stage, we use the same optimizer with different learning rate. We set the learning rate to 0.005 for the 1-shot setting and 0.05 for the 5-shot setting. $\alpha$ is set to 0.25 and 2 for the 1-shot and 5-shot settings respectively. We fine-tune the weights of the transformation $g$ for 20 epochs. Note that all the hyper-parameters are determined by the performance on the validation set of each dataset.

### 4.3 COMPARISONS WITH THE STATE-OF-THE-ARTS

**Results on mini-ImageNet.** Table 1 shows a comparison of our method to current state-of-the-arts on the mini-ImageNet dataset. Classification accuracies are reported for two backbone networks, i.e. Conv-4 and ResNet-12. For Conv-4 backbone, we achieve the best accuracy in both 5-way 1-shot and 5-way 5-shot settings. Our method boosts the performance for the 5-shot setting, surpassing the second best by 9.33% and 2.10% for the Conv-4 and ResNet-12 backbones respectively. A closer analysis on the larger improvement for the 5-shot setting than 1-shot setting is given in the discussion section.

**Results on CUB-200-2011.** On this dataset, we find that there is no work reporting the few-shot performance for the ResNet-12 backbone. For a fair comparison, we report the accuracies of the Conv-4 and ResNet-10 backbones. As shown in Table 2, similar to mini-ImageNet, our method outperforms the second best method by a large margin (87.92% vs. 81.90% for Conv-4 and 92.89% vs. 87.45% for ResNet-10) in the 5-way 5-shot setting. In the 1-shot setting, we achieve the best performance for ResNet-10 and comparable accuracy to PABN (65.11% vs. 66.71%) for Conv-4. It is worth mentioning that in PABN, the max-pooling layers of the backbone network are replaced with bilinear pooling, which is specially optimized for fine-grained classification tasks. We expect further accuracy improvement by incorporating such techniques in our method.

**Results under Domain Shift.** To evaluate the generalizability of the task-agnostic feature learned with the proposed Metric-Softmax loss, we perform the cross domain experiments. In the cross

Table 2: Few-shot classification results on the test set of the CUB-200-2011 dataset.

| Method | Backbone | 5-way 1-shot | 5-way 5-shot |
|--------|----------|--------------|--------------|
| MatchingNet (Vinyals et al., 2016) | | 61.16±0.89 | 72.86±0.70 |
| ProtoNet (Snell et al., 2017) | | 51.31±0.91 | 70.77±0.69 |
| MAML (Finn et al., 2017) | | 55.92±0.95 | 72.09±0.76 |
| RelationNet (Sung et al., 2018) | | 62.45±0.98 | 76.11±0.69 |
| PABN (Huang et al., 2019) | Conv-4 | **66.71±0.43** | 76.90±0.21 |
| Baseline++ (Chen et al., 2019) | | 60.53±0.83 | 79.34±0.61 |
| DN4 (Li et al., 2019) | | 53.15±0.84 | 81.90±0.60 |
| **FTA (Ours)** | | 65.11±0.65 | **87.92±0.38** |
| Baseline++ (Chen et al., 2019) | | 69.55±0.89 | 85.17±0.50 |
| SubspaceNet (Devos & Grossglauser, 2019) | ResNet-10 | 72.92±0.90 | 87.45±0.48 |
| **FTA (Ours)** | | **73.19±0.62** | **92.89±0.28** |

Table 3: Identical domain and cross domain results for 5-way 5-shot classification using the ResNet-10 backbone. The dataset names (mini-ImageNet and CUB-200-2011) are abbreviated to mini and CUB respectively for brevity.

| Method | Identical Domain | | Cross Domain | |
|--------|------------------|--------------|--------------|--------------|
| | mini-ImageNet | CUB-200-2011 | mini → CUB | CUB → mini |
| MatchingNet | 69.14±0.69 | 83.75±0.60 | 52.59±0.71 | 48.95±0.67 |
| ProtoNet | 73.77±0.64 | 85.70±0.52 | 59.22±0.74 | 53.58±0.73 |
| RelationNet | 69.97±0.68 | 82.67±0.61 | 54.36±0.71 | 45.27±0.66 |
| SubspaceNet | 74.03±0.68 | 87.45±0.48 | 62.71±0.71 | 56.66±0.68 |
| **FTA (Ours)** | **86.44±0.36** | **92.89±0.28** | **85.38±0.45** | **58.35±0.52** |

domain scenario, we swap the training sets of the two datasets. Specifically, in the mini-ImageNet→CUB-200-2011 setting, the training set is comprised of the 64 base classes from mini-ImageNet, while the validation and test data come from CUB-200-2011. Similarly, in the CUB-200-2011→mini-ImageNet setting, we use the 100-class base data of CUB-200-2011 for general feature learning and evaluate its performance on the mini-ImageNet dataset. We report the results under the same setting as Devos & Grossglauser (2019) in Table 3. Compared with existing methods, our approach achieves the highest accuracy in all settings. Notably, we improve SubspaceNet by 22.67% in the mini-ImageNet→CUB-200-2011 setting. The strong cross domain performance is mainly attributed to the task-adaptive transformation $g$ of our FTA. If it is disabled, the accuracy drops drastically from 85.38% to 67.22%. Another observation is that the performance degrades more in CUB-200-2011→mini-ImageNet than mini-ImageNet→CUB-200-2011. This is because mini-ImageNet contains more abundant classes than CUB-200-2011 and the network can learn more general and discriminative feature.

## 5 DISCUSSION

**Ablation Study of FTA.** As shown in Table 4, compared with the Softmax classifier, the proposed Metric-Softmax classifier improves the accuracy by 4.77% and 3.41% in the 5-way 1-shot and 5-way 5-shot settings respectively. This improvement is attributed to elimination of the discrepancy between training and inference brought by our Metric-Softmax classifier. By incorporating the task-adaptive transformation, the performance is further boosted, especially in the 5-shot setting.

**TAT vs. Fine-tuning.** To leverage the cue from the supported set, an alternative method is to fine-tune a re-initialized Metric-Softmax classifier similar to Chen et al. (2019). Figure 2 shows a comparison of the proposed task-adaptive transformation to the alternative fine-tuning scheme. In terms of convergence speed, TAT takes only 5 epochs to get an appreciable performance. However,

Table 4: Ablation study of FTA on the mini-ImageNet dataset using the ResNet-12 backbone.

| Method | Metric-Softmax | TAT | 5-way 1-shot | 5-way 5-shot |
|---|---|---|---|---|
| Softmax Classifier | × | × | 48.62±0.56 | 69.99±0.57 |
| Metric-Softmax Classifier | ✓ | × | 53.39±0.55 | 73.40±0.47 |
| FTA | ✓ | ✓ | 58.03±0.58 | 80.73±0.44 |

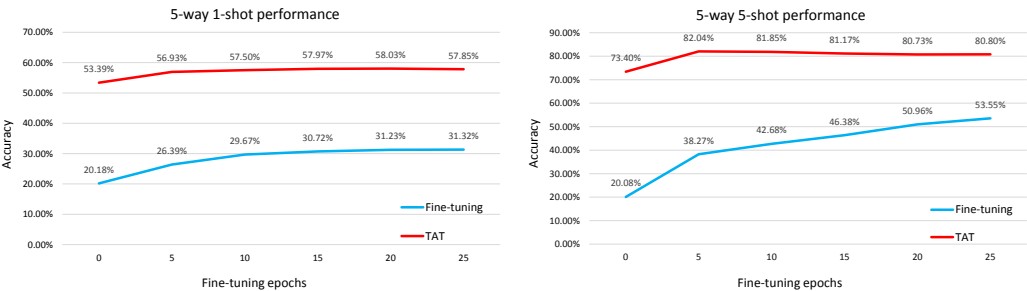

Figure 2: Performance comparison of TAT to direct fine-tuning on the mini-ImageNet dataset using the ResNet-12 backbone. Accuracies on the test set at different training epochs are plotted.

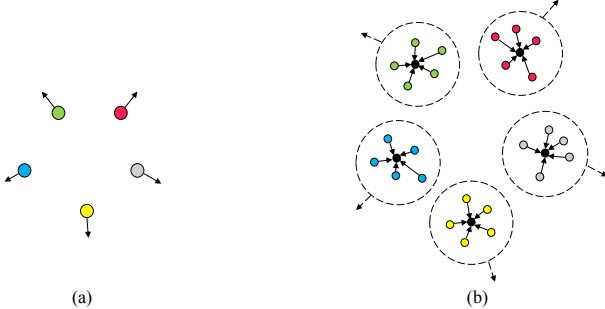

Figure 3: Schematic illustration of different feature transformations by TAT in the 5-way 1-shot (a) and 5-way 5-shot (b) settings. Circles of different colors represent the features of different classes. In (b) the black circles indicate the centroid of each class.

direct fine-tuning takes over 40 epochs to converge for the 5-way 5-shot setting. For the 5-way 1-shot setting, it barely improves the accuracy. In terms of final performance, TAT is far more superior over direct fine-tuning.

**Why does FTA improve the 5-shot setting more than 1-shot setting?** As shown in Section 4.3, our FTA outperforms existing methods more significantly in the 5-shot setting than 1-shot setting. It can be explained by the mechanism of the task-adaptive transformation. In the 1-shot setting, only one image is available in each class. Then Equation 9 degenerates to $\bar{\boldsymbol{W}}_{:,i} = g(\boldsymbol{h}_{(i,1)})$. For an image with label $y, y \in \{1, 2, 3, 4, 5\}$, the numerator of Equation 10 becomes a constant: $\exp(-\alpha||g(\boldsymbol{h}_{(y,1)}) - g(\boldsymbol{h}_{(y,1)})||) = 1$. Now minimizing the cross-entropy means minimizing the denominator of Equation 10, which is equivalent to maximizing the total distance of this feature to others: $\sum_{j}^{5} ||g(\boldsymbol{h}_{(y,1)}) - g(\boldsymbol{h}_{(j,1)})||$. As illustrated in Figure 3(a), task-adaptive transformation only enlarges the distance among different classes in the 1-shot setting. However, for the 5-shot setting (see Figure 3(b)), the transformation not only enlarges inter-class distance but also shrinks the intra-class distance, which helps to improve the classifier.

## 6 CONCLUSIONS

In this paper, we propose the Metric-Softmax loss which can adequately explore the feature similarities in the whole base dataset with the premise of keeping the classifier consistent between training and inference. In addition, we take full advantage of the few labeled novel data by introducing a parameterized transformation to adapt the learned general feature to each few-shot classification task. Our approach is particularly helpful for the 5-shot setting. We leave further improvement for the 1-shot setting as our future work.

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
