# OpenReview forum: "Fast Task Adaptation for Few-Shot Learning"
_ICLR.cc/2020/Conference — Reject_

### Official Review · AnonReviewer1 · 2019-10-15
**Official Blind Review #1**

**Rating:** 8

**Review:**

Authors propose a new method for adaptation in a few-shot learning setting. Their method comprises two different steps; first they propose a new metric-softmax loss, which aims at improving the transferability of features pre-trained on base data to novel data. They achieve this via redefining the probability score calculating function, which in practice means they replace the exponent term found in the softmax loss with a Gaussian kernel-based radial basis function. This first step improves the feature learning process at large scale but does not solve the problems found when trying to fit arbitrary novel classes. At this point comes step two, where a fast task adaptation process is proposed, i.e. Task-Adaptive Transformation based on affine transformation. This method converges fast and is learnt from the support set, vs step 1 which is trained on the training/base set. Post-training the affine transformation is applied to both support and query image sets.
Authors have been fairly detailed in their experimental processes and used different backbone models, routinely found in papers focusing on similar issue. They compare against 4 other state-of-the art methods showing a significant improvement across all of them. They also demonstrate the superiority of the metric-softmax classifier vs softmax-classifier and finally the overall superiority of the whole method proposed.
Similar results are obtained on domain sift settings as well as when comparing the step 2 of this method with a fine-tuning approach.
The proposed method makes a good contribution towards reducing the problem of overfitting when very few examples of a new problem are available.
I have read the rebuttal and think the paper could be a good addition to the programme.

**Experience Assessment:**

I have published one or two papers in this area.

**Review Assessment: Checking Correctness Of Derivations And Theory:**

I assessed the sensibility of the derivations and theory.

**Review Assessment: Checking Correctness Of Experiments:**

I carefully checked the experiments.

**Review Assessment: Thoroughness In Paper Reading:**

I read the paper thoroughly.

---

> ### Author Response · Authors · 2019-11-12
> **Revised paper available**
>
> Thanks for your affirmative comment. Note that we have revised our paper to improve its quality and convincingness. Should you have any concerns, please check out the latest version and refer to our responses to the other reviewers.

---

### Official Review · AnonReviewer3 · 2019-10-25
**Official Blind Review #3**

**Rating:** 1

**Review:**



# Summary

This paper deals with few-shot learning from a metric-learning perspective. The authors propose replacing the softmax loss, i.e. softmax + cross-entropy loss, with a so-called "metric-softmax" loss which imitates a Guassian kernel RBF over class templates/weights. This loss is used in both stages of training on base and on novel classes and the authors argue that it helps learning more discriminative feature while preserving consistency between train and test time.
Secondly the authors advance a task-adaptive transformation for stage 2 that maps the features from the previously learned feature extractor to a space which is easier to learn. The contributions are evaluated on the standard mini-ImageNet benchmark and on CUB-200-2011 individually and in domain shift mode.

# Rating
Although some of the results in the paper might look impressive, my rating for this work is reject for the following reasons (which will be detailed below):
1) the main contribution, metric-softmax loss, is not novel. It has been used and described in multiple works in the past 1-2 years.
2) a part of the evaluations and comparisons do not follow the usual protocol and are not fair
3) the second contribution, Fast Task Adaptation (FTA), is not well described and it's unclear in what it actually consists, how does it work and how was it trained exactly.

# Strong points
- This paper deals with a highly interesting and relevant topic for ICLR.

# Weak points

## Contributions
- This work ignores a large body of research in few-shot learning and metric learning aiming to improve the efficiency per training sample and feature discrimination.
The proposed loss can be traced back to Goldberger et al. [i] in  NCA (Neighborhood Component Analysis). Prototypical Networks are derived from this work and hence similar with the metric-softmax loss.
Qi et al. [ii] point out that when h and W are l2-normalized (using the notations from this submission, eq. 7) maximizing their inner-product or cosine-similarity is equivalent to the minimization of the squared Euclidean distance between them. This leads to the loss from [ii], [iii], also known as Cosine Classifier and which also is accompanied by a scaling factor or temperature as here.
Other related works on improving softmax and selecting the representative weights for a class include: center loss [iv], ring loss[v], L-GM loss[vi].
In this light, the metric-softmax is actually not novel and can be found in several other contributions from last year.

- In my opinion, it is difficult from the paper to understand what is actually the fast adatpation module. The authors describe $g$ as "simply a zero-offset affine transformation" $g(h)= M^T h$. In the implementation details we do not find out more about this module and we don't have more insights on what is it doing inside, other than a toy hand-drawn example in Figure 3. I find it difficult to assess.

## Experiments
- The authors evaluate 3 backbone architectures, Conv-4, ResNet-10 and ResNet-12. For the former they use 84 x 84 images, while for the latter they use 224 x 224 images.  The larger images are not standard in the few-shot ImageNet evaluation protocol. Data augmentation (jittering, flipping, etc.) is used here, while in most works it is not. Chen et al. are the first ones to introduce larger images and data augmentation and acknowledge that the large scores are due to this.
Testing out new configuration is not a problem as long as the baselines are evaluated in the same conditions. However, in this case they are not and this is not visible in the captions of the tables and descriptions in the paper. Training a network with data augmented images and/or higher resolution images and comparing to baselines without data augmentation and images with 7 times less pixels, for sure does not allow seeing the true impact of the proposed method. I would advise to either evaluate in the usual mini-ImageNet settings, either implement a few representative and easy to train baselines, e.g. ProtoNets, Cosine Classifier[iii] in the same conditions as here and compare against. This should provide a better idea on the effectiveness of the proposed methods.


## Other comments
- the scores for baseline methods are seemingly taken from the paper of Chen et al. who trained them themselves. This should be mentioned in the paper and in the caption

# Suggestions for improving the paper:
1) Review the experimental section and make sure at least some of the baselines are trained in similar conditions as the proposed method or alternatively evaluate the proposed methods in standard mini-ImageNet settings

2) Provide additional insights, experiments and implementation details for FTA to make it easier to understand, there are some examples in the references below.



# References
[i] J. Goldberger et al., Neighbourhood components analysis, NIPS 2005
[ii] H. Qi et al., Low-Shot Learning with Imprinted Weights, CVPR 2018
[iii] S. Gidaris and N. Komodakis, Dynamic Few-Shot Visual Learning without Forgetting, CVPR 2018
[iv] W. Wen et al., A Discriminative Feature Learning Approach
for Deep Face Recognition, ECCV 2016
[v] Y. Zeng et al., Ring loss: Convex Feature Normalization for Face Recognition, CVPR 2018
[wi] W. Wan et al., Rethinking Feature Distribution for Loss Functions in Image Classification, CVPR 2018

**Experience Assessment:**

I have published one or two papers in this area.

**Review Assessment: Checking Correctness Of Derivations And Theory:**

I carefully checked the derivations and theory.

**Review Assessment: Checking Correctness Of Experiments:**

I carefully checked the experiments.

**Review Assessment: Thoroughness In Paper Reading:**

I read the paper at least twice and used my best judgement in assessing the paper.

---

> ### Author Response · Authors · 2019-11-13
> **On the novelty of our approach and experimental settings**
>
> Thanks for your comments and suggestions. We have revised our paper as noted in this comment (https://openreview.net/forum?id=ByxhOyHYwH&noteId=HyekWIfdjS ). We believe that the quality and convincingness of our work have been significantly improved. To address your concerns, in particular on the novelty of our approach and setting of experiments, see our explanations below.
>
> ## Contributions of this work
>
> - Metric-Softmax Loss
> What motivate us to design the Metric-Softmax loss are to 1) enjoy the strong feature learning capability of the Softmax loss and 2) preserve the consistency between training, fine-tuning and inference. Although it shares similar form with existing losses, the key to the effectiveness of our approach lies in the overall learning pipeline, rather than the loss alone. The ablation studies in the Section 5 clearly verify the positive impacts brought by both Metric-Softmax and FTA. In other words, we propose a complete learning framework, which is a composite of multiple techniques applied to the training and adaptation stages.
>
> - Fast Task Adaptation
> By transforming the features of novel images using the matrix $\mathbf M$, we aim to learn a more compact and discriminative feature space for each new few-shot classification task. It can be easily implemented. Given d-dimensional feature vectors $\mathbf h$, it is analogous to apply a fully connected layer to $\mathbf h$. It should be noted that 1) the weight matrix $\mathbf M$ is a square matrix such that the output dimension of $\mathbf h$ is unchanged; 2) no bias is applied; 3) $\mathbf M$ is initialized with an identity matrix. This design has two merits. On one hand, the strong features learned on the base data can be maximally preserved. On the other hand, tuning $\mathbf M$ on the support images of novel classes further strengthens the feature to current few-shot task, without suffering from overfitting as a naïve fine-tuning. The transformation is applied to both support and query images, which is distinct from existing works. Please see our responses to Reviewer #2 (A1, B4 and B7) for more detailed analysis.
>
> ## Improved experiments
>
> - The setting that matters most for performance is the input image size. Originally we followed the setting of ResNet-10 in Chen et al. (2019), which turns out to be unusual for ResNet-12 in the literature. This issue has been fixed by rerunning the experiments of ResNet-12 with the 84x84 setting. The latest results have been updated in current revision. On mini-ImageNet the accuracies are 58.03±0.48 and 80.73±0.44 for the 5-way 1-shot and 5-way 5-shot settings respectively, which are still competitive to existing state-of-the-arts.
>
> ## Other issues
>
> - Q: The scores for baseline methods are seemingly taken from the paper of Chen et al. who trained them themselves. This should be mentioned in the paper and in the caption.
> A: Fixed by adding a note in the caption. Thanks for pointing this out.
>
> - More state-of-the-art methods have been added for comparison. See Table 1 for the latest benchmark.

---

### Official Review · AnonReviewer2 · 2019-10-30
**Official Blind Review #2**

**Rating:** 3

**Review:**

This paper develops a new few-shot image classification algorithm. It has two main contributions. The first one is to use a metric-softmax loss used to train on the meta-training dataset without episodic updates. The second is that the features learnt thereby are further modified using a linear transformation to fit the few-shot training data and the metric soft-max loss is again used for classifying the query samples. The authors provide experimental results for 5-way-1-shot and 5-way-5-shot testing on mini-Imagenet and CUB-200-2011 datasets.

I think this paper is below the acceptance threshold. The reasons are:

1. The contributions of this paper are marginal: both learning centroids for each meta-training class and projecting the few-shot features have been used before in published work (https://arxiv.org/abs/1905.04398). The empirical results are weaker than existing work (see for instance, https://arxiv.org/abs/1904.03758, https://arxiv.org/abs/1909.02729 etc.); also see #3 below.
2. The authors should provide experimental results on other few-shot learning datasets like tiered-Imagenet.
3. The image-size used here for Resnet-12 is 224x224, the authors should report results using 84x84 image size so that one can compare against existing literature fairly. Are the results for Resnet-12 so good because of the larger image size?
4. The training procedure is task-agnostic, why do you train a different model for the 1-shot and the 5-shot case?

I will consider increasing my score if some of the concerns above are addressed. I am listing some more comments below which I would like the authors to consider.


1. Contributions: “consistency between training and inference”, do you instead mean consistency between meta-training and few-shot training? There are no weight updates at inference time.
2. How essential is the metric-softmax loss? Training on the meta-training dataset without episodic updates has also been done in https://arxiv.org/abs/1909.02729. These authors seem to use standard soft-max training and perform standard fine-tuning, they report empirical performance that is significantly better than that in Table 4 and Figure 2. I am very skeptical as to why the accuracy of fine-tuning is only 21% in Figure 2.
3. Section 3.2 does not motivate or explain the metric-softmax loss. Why should one have the network learn the centroids of the meta-training dataset? Can you draw a TSNE of the centroids learnt during meta-training? The features of the support samples (or their transformations) can be the centroids of the few-shot classes in the prototypical loss so inference phase does not need these centroids.
4. I am not sure whether the matrix M is changed non-trivially during few-shot training. The weights W are already initialized to be the centroid of the features (eqn. 9). So the metric-softmax loss in eqn. 10 is expected to be small for the support samples after initialization. Why should the additional expression power afforded by M matter? There is no incentive for the network to change the matrix M. Can you show results on how much M changes from the identity?
5. I believe the reported numerical results for LEO (Rusu et al. 2019) are for a WRN-28-10 architecture, not ResNet-12.
6. The accuracy using Resnet-12 seem extremely high. I believe this is because the results reported in the literature, e.g., https://arxiv.org/abs/1904.03758, use images of size 84x84, not 224x224 as the authors here have used. Can you report results using 84x84 sized images?
7. I don’t understand the explanation at the end of Section 5. Since the prototypical loss is being used to classify the query datum, it should not matter whether the cluster is shrunk in the 5-shot case, or whether simply the distances between the clusters are increased as in the 1-shot case.
8. Table 1 is quite incomplete, the authors should mention other existing few-shot classification results are similar to the performance of this paper, e.g., https://arxiv.org/abs/1805.10123, among the ones listed above.
9. The entries in Table 1 and 2 are not made bold appropriately. All entries with overlapping standard error should be bold.

**Experience Assessment:**

I have published one or two papers in this area.

**Review Assessment: Checking Correctness Of Derivations And Theory:**

I carefully checked the derivations and theory.

**Review Assessment: Checking Correctness Of Experiments:**

I carefully checked the experiments.

**Review Assessment: Thoroughness In Paper Reading:**

I read the paper thoroughly.

---

> ### Author Response · Authors · 2019-11-12
> **Responses to reviewer #2**
>
> Thanks for your thoughtful comments. Based on your suggestions, we have carefully revised our paper (https://openreview.net/forum?id=ByxhOyHYwH&noteId=HyekWIfdjS ). In particular, all experiments of ResNet-12 have been rerun with the new input image size (84x84) and the accuracy numbers have been updated in the latest version of the paper.
>
> To further address your concerns, here are our explanations.
>
> A1. As you mentioned, the most related work to ours is Ravichandran et al. (2019) (https://arxiv.org/abs/1905.04398 ), which also learns the class representations followed by projecting the features. The major difference is that they apply the projection to the representative features derived from the support images only. While we apply the affine transformation g to both features of support images and query images. That is, we aim to learn a more compact and discriminative feature space for the full set of images of novel classes, rather than merely adjusting the class representations of novel classes. This is the key to explain the observation that we improve the 5-shot setting more than the 1-shot setting. When there is 1 image per class, it is difficult to learn a class-level transformation given single image-level information. When there are multiple (e.g. 5) images per class, it is able to learn a class-level transformation that generalizes well to query images. This analysis is well supported by the latest results in the updated Table 1, where we achieve higher accuracy (80.73% vs. 77.46%) than Ravichandran et al. (2019) in the 5-shot setting.
>
> A2. We believe that reporting results on two datasets using three network backbones is fairly convincing. We will add experiments on more datasets in the future revision.
>
> A3. The input image size of ResNet-12 has been changed from 224x224 to 84x84 for a fair comparison with existing methods. And all the related experiments have been rerun using the new setting. The accuracy numbers have been updated accordingly (see Table 1, Table 4 and Figure 2). Originally we followed the setting of ResNet-10 in Chen et al. (2019), which turns out to be unusual for ResNet-12 in the literature. We apologize for the confusion, and would like to thank the reviewers and readers for pointing this issue out.
>
> A4. The 1-shot and 5-shot settings share the same backbone model. During evaluation, since different support and query image sets are sampled randomly in different episodes, we fine-tune the transforming matrix M for each episode independently, leading to different Ms for different few-shot classification tasks. In other words, the training procedure is identical for all tasks, what differs is the list of support images used for fine-tuning.
>
>
> B1. The consistency between training and inference lies in the way to compute the class probability. In Softmax, the probability is computed by vector inner-product in training and Euclidean distance in inference. In Metric-Softmax they are all computed by Euclidean distance.
>
> B2. The reason is that we used the same scaling factor $\alpha$ (0.25) for both TAT and fine-tuning. It works perfectly fine for TAT, but is too small for fine-tuning as the weights are randomly re-initialized in fine-tuning. In the revised version, we set $\alpha$ to 15, which is the same as training on the base data, and obtain a more reasonable result (31.32% at the 25-th epoch).
>
> B3. As explained in B1, we design the Metric-Softmax loss to ensure the consistency between training and inference. During meta-training, the weight matrix W is learned from scratch. It can be interpreted as the centroids (or representations) of base classes only due to the definition of the classifier in Eq. (7). During fine-tuning, it can not be transferred to novel classes. Thus we use the centroids of novel classes as the initial weight for the classifier on novel classes.
>
> B4. By inspecting M, we found that indeed the change is trivial and it is close to the initial identity matrix. The diagonal elements of M range from 0.94 to 1.05, while the rest elements are of the magnitude of 1e-2. It indicates that a slight transformation to the base features already suffices.
>
> B5. Yes, you are right. It has been fixed in current revision.
>
> B6. We have rerun the experiments of ResNet-12 using the conventional 84x84 setting. The latest accuracies on mini-ImageNet are 58.03±0.48 and 80.73±0.44 for the 5-way 1-shot and 5-way 5-shot settings respectively, which are still competitive to other state-of-the-arts (see the updated Table 1).
>
> B7. Suppose the features of query images are fixed, it does not matter whether the cluster is shrunk. However, the transformation matrix is applied to both query and support features during inference. Therefore, learning a compact feature representation by minimizing the intra-class distance is helpful. Please refer to A1 for more detailed analysis.
>
> B8. More state-of-the-art methods have been added for comparison in the revised paper.

---

### Public Comment · ~Ning_Ma1 · 2019-10-06
**Question About Inference Time**

Hi,I notice that on the Fast Task Adaptation Stage,you use the support data of NOVEL class images .Is it mean that you use test  data (Novel class data) to train classifier  in the inference time?
May be I don't totally understand what the framework do in the inference time.

---

> ### Author Response · Authors · 2019-10-08
> **How we split the dataset and fine-tune on the support images only**
>
> Hi Ning,
>
> Thanks for your attention to our work. In a common few-shot learning setting like mini-ImageNet, both validation and test sets are drawn from novel classes unseen in the training set. This is distinct from normal supervised learning like the full ImageNet-1k classification. In Ravi & Larochelle (2017) which we followed, the three splits are referred to as meta-training, meta-validation and meta-test sets. And in each meta-set, the support and query images are referred to as training and test sets respectively. The support images in the novel data are meant to provide a hint for classification of the query images. Testing accuracy is measured on the query images only. In existing works, the support images have been exploited to learn a nearest-neighbor classifier (Ravi & Larochelle, 2017) or fine-tune a Softmax classifier (Chen et al., 2019). In our framework, we only utilize the support images (NOT query images) to fine-tune the transforming matrix, which is both conventional and reasonable.
>
> References
> Sachin Ravi and Hugo Larochelle. Optimization as a model for few-shot learning. In ICLR 2017.
> Wei-Yu Chen, Yen-Cheng Liu, Zsolt Kira, Yu-Chiang Frank Wang, and Jia-Bin Huang. A closer look at few-shot classification. In ICLR 2019.

---

> > ### Public Comment · ~Ning_Ma1 · 2019-10-09
> > **Compare Inference Stage With Others**
> >
> >     In the paper "A closer look at few-shot classification", they test average accuracy over 600 tasks(episodes).And for each task's adaptation stage , they use support set to train(fine-tune) a new classifier for 100 iteration steps (see section 4.1) .
> >    In your paper,you test average accuracy  over 1200 episodes(tasks).But in the task adaptation stage,you fine-tune transformer g  for 20 epochs,which is very important  differing with the former work.In general understanding,every epoch includes many tasks. 20 epochs might let the neural network traversal the whole test data set several times.  I mean that the transformer g might remember the test data with so many adaptation epochs.
> >   On the other hand,is 'epoch' equal to 'step' in your paper?

---

> > > ### Author Response · Authors · 2019-10-09
> > > **Clarification of fine-tuning in the task adaptation stage**
> > >
> > > For each randomly sampled episode, we fine-tune the transformer g using the support images of current episode only, rather than the whole test set. That is, g is not shared among different episodes, and a unique g is learned for each episode. Fine-tuning is performed strictly on the training (support) data of each task, and it is impossible for g to see the test query images.
> > >
> > > During fine-tuning, we use a batch size of 4. Thus 1 epoch involves 2 iteration steps for the 5-way 1-shot setting and 7 steps for the 5-way 5-shot setting. We will clarify this in the revision.

---

> > > > ### Public Comment · ~Ning_Ma1 · 2019-10-09
> > > > **Thanks for your clarification**
> > > >
> > > > Thanks for your clarification.

---

### Public Comment · ~Bin_Liu4 · 2019-10-13
**Question about Adam optimizer.**

Hi, in the section 4.2, you said you use Adam optimizer with $\epsilon = 10^{-3}$. Is the $\epsilon$ learning rate or the $\epsilon$ parameter in the original paper which is used to improve numerical stability?

---

> ### Author Response · Authors · 2019-10-14
> **It should be the initial learning rate**
>
> Oops, it is a typo. Thanks for pointing out! It should be the initial learning rate rather than Adam's eps parameter, for which the default value (1e-8) of PyTorch's implementation is used. We will fix it in the revision.

---

### Public Comment · ~Cantona_ViVian1 · 2019-10-13
**Question about Fine-tuning on miniImageNet**

The performance of fine-tuning on 5-way 1-shot in Figure 2 is very surprising. It means fine-tuning does not work at all in this case. However, a fixed feature extractor can still get a reasonable accuracy in this case. One possible reason is overfitting. But it is still hard to explain why the performance at the 5th or 10th epoch is very low. Could the authors explain the possible reasons? Thanks.

---

> ### Author Response · Authors · 2019-10-14
> **Issues that make fine-tuning difficult to improve the classifier**
>
> The proposed TAT differs from direct fine-tuning in two aspects. One is initialization of the Metric-Softmax classifier's weight matrix $\mathbf W$. For direct fine-tuning, it is re-initialized randomly. While for TAT, it is derived from features of the support images, which makes it equivalent to a nearest-neighbor classifier. That is why the accuracy starts from random guessing (19.95%) for fine-tuning and a reasonable value (58.76%) for TAT. Better initialization leads to better performance. The other difference is that rather than learning the weights directly as in the case of fine-tuning, TAT learns the transforming matrix g. It is initialized with an identity matrix, which further eases the optimization. In summary, poor initialization and ill-posed learning manner make direct fine-tuning difficult to improve the classifier. These issues emerge in the 1-shot setting, and can be alleviated with more training samples as shown in the 5-shot setting.

---

> > ### Public Comment · ~Cantona_ViVian1 · 2019-10-14
> > **Further questions**
> >
> > I noticed that a fixed feature extractor can works pretty well in the 1-shot case. Based on your reasoning, fixed feature extractor should suffer from the same problem as fine-tuning?

---

> > > ### Author Response · Authors · 2019-10-15
> > > **Key factors to the success of the proposed framework**
> > >
> > > Yes, you are right. If we classify a query image by comparing its feature with features of support images, we may still get a reasonable accuracy, thanks to the generalizability of the feature extractor learned on large-scale base data. In the proposed TAT, we initialize the weights of the Metric-Softmax classifier using the mean features of support images as stated in Section 3.3 (Eq. (9)). In this way, we easily inherit a good starting point. And by fine-tuning the task-adaptive transformation g, further improvement is obtained, as shown in Figure 2. In other words, the combination of the proposed Metric-Softmax classifier together with TAT circumvents the problem of overfitting on extremely few samples, which is one of the major contributions of this work.

---

> > > > ### Public Comment · ~Cantona_ViVian1 · 2019-10-15
> > > > **Re: Key factors to the success of the proposed framework**
> > > >
> > > > Thanks for your clarification. It is still not clear why a fixed feature extractor can work pretty well in the 1-shot case but directly fine-tuning cannot.

---

### Public Comment · ~Alex_Matthew_Lamb1 · 2019-10-14
**Another baseline that perhaps should be added or cited**

At the very least would be good to add the results to the table.  It's not my paper, but it seems like your results are often (but not necessarily always) better:

https://arxiv.org/abs/1907.12087

---

> ### Author Response · Authors · 2019-10-15
> **Different network backbones are used**
>
> Thanks for your reminder. But the paper you mentioned uses different network backbones from ours. We report the performance of Conv-4, ResNet-10 and ResNet-12, while they use ResNet-18, ResNet-34 and WRN-28-10, which are deeper than ours and less commonly used in previous works. As shown in Chen et al. (2019), size of the backbone matters. And the results of different backbones can not be directly compared. Nevertheless, we will include it as a related work in the revision.

---

### Public Comment · ~Jinghan_Gao1 · 2019-10-15
**How is the matrix M trained in stage 2?**

All parts are clear in the paper, except the training process of layer g, which is the essential part of the fast adaptation session. Would you mind explaining the process in detail? I am confused that you only mentioned g is trained with CrossEntropy loss. Do you have a classifier that decreases the dimension of feature while training g?

---

> ### Author Response · Authors · 2019-10-16
> **The detailed learning process of fast task adaptation**
>
> To answer your question in short, we learn g using the (modified) Metric-Softmax loss.
>
> Training of g is analogous to training of the feature extractor using the Metric-Softmax loss, which aims to minimize the divergence between predicted classification scores to one-hot labels using the cross-entropy loss. The differences lie in
>   1) the classifier to predict the scores, i.e. Eq. (10) vs. Eq. (7),
>   2) the weights to update, i.e. updating g only vs. updating the whole network,
>   3) the number of classes in the classifier, e.g. 5 vs. 64 for miniImageNet.
> This consistency of learning process is one of the factors that contributes to the performance improvement.
>
> The transformation g is applied to the extracted features (Eq. (8)). Since the matrix M is a square matrix, the dimension of the feature is kept unchanged.

---

### Author Response · Authors · 2019-10-16
**More implementation details that may affect the reproducibility of this work**

To the reviewers and readers,

By double-checking the implementation after submission of the paper, we noticed a few missing details that may affect reproducibility of this work.

1) In the Metric-Softmax classifier (Eq. (7)), the feature vector $\mathbf h$ should be L2-normalized, which eases optimization in our experiments.
2) Analogously, in the modified Metric-Softmax classifier used during the fast task adaptation stage (Eq. (10)), both the transformed feature vector $g(\mathbf h)$ and each column of the weight matrix $\bar{\mathbf W}$ should be L2-normalized.
3) For the scaling factor $\alpha$ in Metric-Softmax, we use different values in the two stages. In the feature learning stage, it is set to 15 and 1 for mini-ImageNet and CUB-200-2011 respectively. In the fast task adaptation stage, it is set to 0.25 for the 1-shot setting and 2 for the 5-shot setting on both datasets. We would like to stress that these parameters are tuned strictly on the validation set of each dataset.

We apologize for these issues and will fix them in the following revision.

---

### Public Comment · ~Yue_Wang2 · 2019-10-17
**Input image size**

I want to mention that in TADAM, the image resolution they use is 84x84 while in your paper, you use 224x224, right?

---

> ### Author Response · Authors · 2019-10-17
> **Explanation on input image size**
>
> In our work, we use an input size of 84x84 for Conv-4 and 224x224 for ResNet-10 and ResNet-12, which follows the setting of Baseline++ (Chen et al., 2019) and SubspaceNet (Devos & Grossglauser, 2019). TADAM uses an input size of 84x84 for ResNet-12. We include TADAM in the table to make the comparison more comprehensive. To make it clear, we will mark the difference in input size in the revision.

---

### Author Response · Authors · 2019-11-12
**Revised paper uploaded (fixing the wrong input image size issue)**

Dear reviewers and all,

We have uploaded a revised version of our paper to address the major issues raised by the reviewers and readers. The changes can be summarized as follows.

1) The input image size of ResNet-12 has been changed from 224x224 to 84x84 for a fair comparison with existing methods. And all the related experiments have been rerun using the new setting. The accuracy numbers have been updated accordingly (see Table 1, Table 4 and Figure 2). Originally we followed the setting of ResNet-10 in Chen et al. (2019), which turns out to be unusual for ResNet-12 in the literature. We apologize for the confusion, and would like to thank the reviewers and readers for pointing this issue out.

2) More state-of-the-art few-shot learning methods have been added for comparison as suggested by Reviewer #2.

3) The latest accuracies for ResNet-12 on mini-ImageNet are 58.03±0.48 and 80.73±0.44 for the 5-way 1-shot and 5-way 5-shot settings respectively. Compared with the strong baseline MetaOptNet-SVM (Lee et al., 2019), we still achieve better accuracy in the 5-way 5-shot setting, confirming the effectiveness of the proposed approach.

4) All the missing implementation details mentioned in this comment (https://openreview.net/forum?id=ByxhOyHYwH&noteId=H1g2MBMEKr ) have been added.

We believe that the quality and convincingness of the paper have been significantly improved.

---

### Decision · Program_Chairs · 2019-12-19

**Decision:**

Reject

**Comment:**

This paper develops a new few-shot image classification algorithm by using a metric-softmax loss for non-episodic training and a linear transformation to modify the model towards few-shot training data for task-agnostic adaptation.

Reviewers acknowledge that some of the results in the paper are impressive especially on domain sift settings as well as with a fine-tuning approach. However, they also raise very detailed and constructive concerns on the 1) lack of novelty, 2) improper claim of contribution, 3) inconsistent evaluation protocol with de facto ones in existing work. Author's rebuttal failed to convince the reviewers in regards to a majority of the critiques.

Hence I recommend rejection.